# Lead Toxicity and Pollution in Poland

**DOI:** 10.3390/ijerph17124385

**Published:** 2020-06-18

**Authors:** Angelika Edyta Charkiewicz, Jeffrey R. Backstrand

**Affiliations:** 1Department of Public Health, Faculty of Health Sciences, Medical University of Bialystok, 15-295 Bialystok, Poland; 2School of Public Affairs and Administration, Center for Collaboration and the Urban Child, Rutgers University, Newark, NJ 07102, USA; backstjr@newark.rutgers.edu

**Keywords:** lead absorption, lead poisoning, lead’s effects, lead exposure

## Abstract

Background: Human exposure to lead can occur in a variety of ways, all of which involve exposure to potentially toxic elements as environmental pollutants. Lead enters the body via ingestion and inhalation from sources such as soil, food, lead dust and lead in products of everyday use and in the workplace. The aim of this review is to describe the toxic effects of lead on the human body from conception to adulthood, and to review the situation regarding lead toxicity in Poland. Results: Pb is very dangerous when it is absorbed and accumulates in the main organs of the body, where it can cause a range of symptoms that vary from person to person, the time of exposure and dose. Lead in adults can cause an increase in blood pressure, slow nerve conduction, fatigue, mood swings, drowsiness, impaired concentration, fertility disorders, decreased sex drive, headaches, constipation and, in severe cases, encephalopathy or death. Conclusions: Exposure to lead in Poland remains an important public health problem. This review will cover the range of lead exposures, from mild to heavy. Public health interventions and policies also are needed to reduce occupational and environmental exposure to this element.

## 1. Introduction

Lead (Pb) is a potentially toxic element that, when absorbed by the body, accumulates in blood and bones, as well as in organs such as the liver, kidneys, brain and skin. Its negative health effects can be both acute and chronic, because the human body poorly excretes Pb. In humans, lead has been shown to affect the function of the reproductive, hepatic, endocrine, immune and gastrointestinal systems [1]. There is limited evidence of a carcinogenic effect of lead and its inorganic compounds on humans [2].

Human exposure to lead can occur in a variety of ways, all of which involve environmental pollution. Lead enters the body via ingestion or inhalation from sources such as soil, food, lead dust and contact with lead in products of everyday use and in the workplace (Figure 1). In the work environment, the main route of absorption of Pb and its compounds is through the respiratory system, although lead is also absorbed via the digestive system (atsdr.cdc.gov) [3,4,5,6,7]. Lead has had many different industrial applications in the past, and is currently used for a range of purposes. Currently, Pb is used in battery plates and equipment for the production of sulfuric acid, cable covers, soldering materials, shields in atomic reactors, aprons and containers for radioactive materials, in the paint and ceramics, and chemical and construction industries, and in the production of bearings and printing fonts, and aviation gasoline, etc. Until recently, lead-based house paint and tetraethyl lead in gasoline were major sources of environmental lead [2,3,4]. Toxic exposure to lead in Poland remains an important public health problem. Constantly expanding knowledge in the field of pollution by toxic elements, including lead, is particularly important, because they are broadly harmful to health and even human life. In the case of Pb, it is necessary to determine the levels of exposure as well as absorption, as the effects of Pb poisoning are often visible only after a few years.

The aim of this review is to describe the toxic effects of lead on the human body from conception to adulthood. The second main aim is to describe sources and levels of lead contamination in Poland, where exposure to environmental lead varies by region.

## 2. Lead Absorption

### 2.1. Lead Absorption, Excretion and Storage in Body

#### 2.1.1. Ingestion

The ingestion of lead occurs by the consumption of lead-contaminated food and water, and from lead-containing substances which are inhaled and subsequently absorbed (Figure 1). Tetraethyl and tetramethyl lead are absorbed by the digestive system and transported to the blood, from where they are distributed to various tissues within an hour [4,8].

Diet and nutritional status are important factors that can influence lead absorption. A diet rich in zinc, copper, iron or calcium can reduce the absorption of lead by the digestive system. Diets rich in fat or protein can improve overall vitamin D status, which can increase lead uptake. Phosphorus in food can cause the conversion of lead to highly-soluble phosphates, which can be absorbed to a limited extent. Intake of vitamin C with Pb may increase lead absorption. Lead is also capable of displacing essential microelements from metalloenzymes. It can also impede the biosynthesis of heme and cause disturbances in microelement (Fe, Cu, Zn Se and Ca) metabolism [2,9,10,11]. Calcium and phosphorus administered in a single dose reduce Pb absorption from the gastrointestinal tract by 1.3 and 1.2 times, albeit less than when these nutrients are given jointly (which can result in a 6-fold reduction). Zinc also plays a protective role, reducing ALAD inhibition by Pb. Lead displaces zinc readily in one of the alloenzymes of the protein. The relationship between δ-aminolevulinic acid dehydratase genotype and sensitivity to lead at different blood lead concentrations is, at present, unclear. Pb also causes an increase in zinc protoporphyrin by a mechanism that is not fully understood [2,10]. Pb absorption in people with iron deficiency may be 2–3 times higher than in people without iron deficit [9,12,13].

#### 2.1.2. Lead in the Human Body

Following absorption, Pb concentrations in the blood equilibrate approximately 3 months after exposure (atsdr.cdc.gov). Most lead is stored in the liver, and to a lesser degree, in the kidneys; the remaining lead is dispersed throughout the body (cerebral cortex, spinal cord, ovary, pancreas, spleen, prostate, adrenals, brain, fat tissue, testes, heart and skeletal muscles) [14].

In adults, Pb is found in bones, and concentrations increase with age by up to ten times, especially in the tibias. After lead exposure, its elimination comprises two phases: the first (elimination from blood and soft tissues) takes about 20–30 days; the second, the slow elimination phase of lead from the blood, involves excretion from the bones. The biological half-life of lead in trabecular bone is estimated to be one year, and in cortical bone from 10–20 years [15].

Most inorganic lead is removed via the urine (approximately 2/3), while another 1/3 is expelled in bile into the intestine and then removed from the body in feces. Small amounts of lead (Figure 2) can be secreted in sweat, milk and saliva, or can accumulate in the hair and nails (atsdr.cdc.gov) [4,8]. After exposure, the lead elimination half-life from blood and soft tissues is about 30 days, and from bone 10–20 years; as a result, lead can remain in the body for decades [14,16,17].

### 2.2. Lead’s Effects

#### 2.2.1. Metabolic and Genetic Effects

Lead impairs multiple biochemical processes, including inhibiting calcium and reacting with proteins. Upon entering the body, Pb takes the place of calcium and then interacts with biological molecules, interfering with their normal function. Lead reduces the activity of various enzymes, causing changes in their structure, and inhibits their activity by competing with the necessary cations for binding sites. Oxidative stress caused by lead is the main mechanism responsible for its toxicity, causing changes in the composition of fatty acids in membranes (affecting processes such as exocytosis and endocytosis, and signal transduction processes). Pb can also cause gene expression alterations. Some research has investigated the effects of Pb on the activity of glucose-6-phosphate dehydrogenase (G6PD). It has been shown that by causing anemia, it may interfere with the integrity of the RBC membrane, making it more fragile. Pb can also inhibit the enzyme ferrochelatase, reducing iron (Fe) incorporation into heme. Lead inhibits the δ-aminolevulinic acid dehydratase (δ-ALAD) enzyme, leading to increased blood levels of δ-aminolevulinic acid (δ-ALA). Pb-induced oxidative damage is the result of a disturbance in the balance of glutathione (GSH) to glutathione disulfide (GSSG). The presence of lead in the body can cause rapid depletion of antioxidants in the body, and can increase the production of reactive oxygen, as well as reactive forms of nitrogen. Thus, increased oxidative stress causes a reduction in the levels of glutathione reductase, leading to a reduction in the concentration of the antioxidant glutathione [10,20,21].

#### 2.2.2. Location of Lead in the Body

Most lead is stored in the bones [15,22], where it is not uniformly distributed. It tends to accumulate in bone regions undergoing the most active calcification at the time of exposure. The rates of development and accumulation suggest that accumulation will occur mainly in the trabecular bone during childhood, and in the cortical bone in adulthood. Bone-to-blood lead mobilization increases with age, broken bones, chronic disease, hyperthyroidism, kidney disease, pregnancy and lactation, menopause and physiologic stress. Calcium deficiency exacerbates bone-to-blood lead mobilization in all of the above instances [1,3]. Recently, research in southern Spain identified elevated lead levels in adipose tissue, although the authors noted that further research is needed on this subject [23].

#### 2.2.3. Children

WHO experts stress the importance of Pb control among children, because research consistently shows that it adversely affects the central nervous system and development [24]. Pb is especially harmful to children under the age of six, most likely because of the rapid brain growth and development with associated periods of heightened vulnerability, and because of high demand for nutrients [2,9,24]. Pb can interfere with the ability to learn, impair memory, lower IQ and interfere with growth and development. Pb has documented effects on speech, hearing, hyperactivity, nerve conduction, intestinal discomfort, constipation, vomiting, weight loss and muscle aches. At high blood concentrations (Table 1), lead poisoning can lead to anemia, nephropathy, paralysis, convulsions or death [4,10].

Damage to the child due to lead can begin as early as pregnancy. Maternal lead can be passed through the placenta to the developing fetus [10,18]. The Pb content in the placenta is the result of many complex biochemical reactions and various factors related to the mother’s body. The concentration of Pb in umbilical cord blood can be up to 85% of that in the mother’s blood. When a woman becomes pregnant, the lead stored in her bones can be released and transferred through the blood to the fetus, especially if the mother’s calcium intake is low. Therefore, fetal development can be influenced by both current and past maternal exposure to Pb via lead stored in the mother’s bones [25,26].

Research has found that even slight Pb exposure is associated with increased the risk of miscarriage, stillbirth, low birth weight and underdeveloped children. It is still not known what Pb levels can cause mutations and congenital abnormalities in the fetus, as well as the exact mechanisms of these changes. Based on current evidence, the WHO has targeted blood Pb concentrations of 5 μg/dL or less for children. Damage from lead exposure can occur at levels below this value [2,9,24,25,26]. Research on children’s blood pressure and prenatal lead exposure in Mexico City (assessed by lead concentration in mothers’ tibia) found an association between lead and higher blood pressure in girls, but not in boys [27].

Severe lead poisoning in children can cause dementia, irritability, headaches, muscle twitches, hallucinations, memory disorders, learning or behavioral problems, concentration and attention issues, a reduction in IQ, hearing loss, restlessness or hyperactivity. Acute poisoning can lead to convulsions, paralysis and coma. In fatal cases, brain damage can occur due to edema and changes in the blood vessels [4,8,10].

Children are exposed to environmental lead via inhalation and ingestion. Inhalation contributes to higher blood levels in children than in adults. Dirt, dust and food are the largest sources of Pb in children, while ingestion from water is generally a less significant source (Figure 3). The exposure of young children to lead is substantially enhanced by the common infant and toddler behavior of tasting objects, putting hands into the mouth immediately after playing, etc. [2,10]. When young children live in environments with Pb contamination, the ingestion of potentially toxic elements is likely. Increased risk of lead poisoning occurs in families where one parent works in an environment with high levels of lead. Parents who are exposed to Pb in the workplace often bring lead dust into the home on their clothes or skin, thereby increasing the risk of exposure for their children to workplace lead. The monitoring of Pb concentrations in children’s blood is recommended (atsdr.cdc.gov) because of their vulnerability [2,5,6,9].

#### 2.2.4. Adults

The effects of lead exposure in adults are underappreciated. High lead concentrations can result in serious morphological and functional changes in some organs [2,26]. Lead in adults can cause changes in the nervous system (causing slow nerve conduction, fatigue, mood swings, drowsiness, concentration disorders, headaches, coma), in the circulatory system (increasing blood pressure, and in severe cases, encephalopathy), in the gastrointestinal system (colic/pain, nausea, vomiting, diarrhea and constipation), and hormones (fertility disorders, decreased libido); other effects include astringency of the mouth, metallic taste, and thirst or even death [2]. Additionally, Pb can seriously affect the cardiovascular system. People exposed to very high doses of Pb (blood lead concentrations between 500–870 μg/L blood) can experience sinus node dysfunction, atrioventricular conduction disturbances and atrioventricular block [17]. Pb exposure can also lead to morphological changes in the heart, e.g., visible changes in the electrocardiographic picture, impaired systolic and diastolic function, changes in repolarization dispersion and increased blood pressure [17,21,28].

The relationship between lead concentrations in blood and blood pressure has been widely studied; many researchers have observed a positive relationship, whilst others have not. Any effect of lead on blood pressure would be dependent on the exposure dose and the time of exposure [2]. Even low lead exposure can cause cardiovascular disease [29], is associated with oxidative stress and deficiency of the enzyme catalase, and may contribute to hypertension. Increasing the level of lead in the body causes greater cardiovascular responses to acute stressors. Increased cardiovascular reactivity predicts higher baseline blood pressure, increased left ventricle mass and atherosclerosis in adults. Consequently, increased blood pressure reactions to acute stressors are one of the possible mechanisms by which lead might affect resting blood pressure. Also, lead levels in the blood may affect cardiac output or the total resistance of peripheral vessels, and thus increase the blood pressure response to acute stressors [3,14,30]. Another adverse effect of lead exposure is the effect on the myocytes of the muscular layer of blood vessels. Many researchers have pointed to vasoconstrictor effects in chronic lead intoxication, but this is not firmly established [17]. There is a need to determine the precise role of lead in the pathogenesis of hypertension [28]. A literature review by Navas-Acien et al. concluded that there are causal relationships of lead exposure with hypertension and heart rate variability [31].

Lead compounds can adversely affect blood and the metabolism of blood cells. This red blood cell (RBC) effect manifests in a disorder of cell metabolism of the red blood cell line in the bone marrow or mature erythrocytes. Pb impairs the integrity of the permeability of the membrane, to which RBCs are more susceptible. Heme synthesis is also disturbed by Pb exposure [10].

Lead exposure has been associated with cancer. A meta-analysis assessed the relationship between brain tumors and occupational exposure in several countries (USA, Finland, Sweden, Australia and Russia) [32]. Subsequent research in Finland and the United Kingdom confirmed a relationship between blood lead concentrations and some cancers (brain and lung) [33].

As in children, lead exposure can have adverse effects on the adult nervous system. The blood Pb concentration threshold for asymptomatic CNS function disturbances has been set at 400–600 μg/L (atsdr.cdc.gov) [15]. Adverse effects include impaired visual intelligence, eye-hand coordination and memory, decreased learning ability, impairment of the ability to praise, and potentially, visual and auditory disturbances.

In men who have been exposed to Pb, there can be a reduction in sperm count (>40 μg/dL) and motility, reduced semen quality, morphological disorders, longer time to pregnancy, sterility/impotency and endocrine disorders.

In women, toxic lead levels can lead to miscarriage [18], low birth weight, premature delivery and developmental problems in children. Lead in the mother’s blood passes to the fetus through the placenta and through breast milk [2]. The symptoms of poisoning according to the degree of exposure in children and adults are shown in Table 1.

Lead colic is a frequent result of short-term exposure to large doses of Pb. At onset of symptoms, the person is hungry, and has indigestion and constipation. Following this comes extensive paroxysmal abdominal pain, pale skin and bradycardia. Acute coronary encephalopathy is rare in adults, but when it has occurred, the blood lead concentrations were 800–1000 to 3000 μg/L (atsdr.cdc.gov) [15].

Studies in an adult Chinese population in Wuhan identified associations between lead in urine and diabetes. However, the authors emphasized that more research is needed to confirm this relationship [34].

A summary of the biological effects of lead is provided in Table 2.

### 2.3. Lead Exposure in Poland

#### 2.3.1. In the Environment

Poland is home to some of the richest zinc-lead ore deposits in Europe (the Silesian Uplands and Kraków-Częstochowa Uplands). Mining operations began as early as the 12th century and these regions were leading producers of Pb in the Middle Ages. In prior centuries, Pb was commonly used for covering roofs, window frames, pipes, tableware, jewelry, weights, making glass, shooting balls, printing fonts and for the toy industry for the production of lead soldiers [1,11,35].

In Poland, as in many other countries, lead in gasoline was a major pollutant for much of the 20th Century. Thanks to legislative changes in Poland in the 1990s and in the first decade of the 21st century, a significant drop in the concentration of Pb in atmospheric air occurred, mainly due to the elimination of lead in car fuel [24]. However, since 2010, Pb emissions in Poland have been reduced by just 10% [36]. Forest ecosystems in Polish national parks (concentrations of Pb in two layers of soil: Ojcowsk—2.32 mg/kg, Magurski—1.25 mg/kg, and Gorczański National Parks—0.92 mg/kg) located in the south-west regions of the country have been shown to be polluted, caused by higher industrial activity and the transboundary transport of air pollutants. The most polluted air occurs in the vicinity of mines and metalworks in the Voivodeships: Śląskie, Opolskie and Małopolskie [37].

A special report prepared for Poland in 2016 assessed total emissions of potentially toxic elements in the country: arsenic, chromium, zinc, cadmium, copper, nickel, lead, and mercury. Depending on the types of industrial activity, Pb almost always ranked second in terms of environmental pollution due to the burning of coal. This means that it is dangerous for the environment, and it has very serious consequences for the human health. Pb occupied the first place in waste management and the third in road transport [36].

The work environment is a significant source of lead exposure for many, and leads to numerous health issues and employee absenteeism [38]. Many workers, such as miners and steel workers, welders, plumbers and fitters, car mechanics, glass producers, construction workers, battery manufacturers and recyclers, shooting range employees and plastics manufacturers, are exposed to risk of contact with lead [2]. One study found the average Pb concentration (29.39 ± 17.05 µg/L) in the group of men working in the metal industry in Podlasie Voivodeship was below the biological limit in Poland (270 to 300 μg/L) [38]. Another study found a higher average Pb blood lead concentration (33 ± 9.6 µg/L) in a group of male employees in mills in southern Poland (Śląskie Voivodeship). Other workers who were exposed to lead in mills in Legnica and Głogów had Pb concentrations that were 10 times higher than a group of men working in the metal industry in Podlasie Voivodeship [28,38]. Men who are occupationally exposed to lead should be carefully monitored to preserve their health [38].

#### 2.3.2. In Paints

In Poland, Pb was present in paints manufactured before 1978. Unfortunately, lead paint can still be found on many painted surfaces in older homes, such as doors, windows and verandas. Lead dust and chips can come from paint lead that peels and falls off painted surfaces. The grinding and stripping of lead paint during repainting or renovation can create a serious problem with lead dust, and old paint which has fallen off can also contaminate the soil [1,10]. Currently, lead acetate is found only in some hair dyes [4,35]. Lead-rich inks are currently not used in Poland; however, lead remains in many locations because of its low soil mobility [9].

#### 2.3.3. Lead in Food

An estimated 97% of agricultural soils in Poland are characterized by natural or only slightly increased Pb content [9]. However, in areas under the influence of the steel industry, lead content in soil can significantly exceed the standard by 10 times (OJ 2002, No. 165, item 1359) (Table 3). Public health professionals assume that food grown in soil near active landfills or shipyards may contain dangerous amounts of Pb. Concentrations may depend on the type of parent rock from which the soil was formed, and the pollutants emitted in a given area, e.g., through transport and industry. The absorption of lead by plants depends on the properties of the soil, the characteristic features of the species or the physiological state of the plant (dicotyledons absorb metals much more readily than monocotyledons). Plant roots (including lettuce, radishes, beets, spinach, parsley, carrots, beetroot, white cabbage, cauliflower) can effectively absorb lead, mainly near industrial plants. Among food products, the highest Pb concentrations have been observed mostly in leafy vegetables (0.796 mg/kg) [9,12].

Technological processes are an important source of contamination of food products. The source of lead can be devices used in food production, and can come from various types of dishes, packaging, kitchen appliances, canning, soldered cans and ceramic products such as glassware or porcelain (most often decorative). To reduce this amount, it is necessary to properly prepare the product before consumption, wash thoroughly, and cook or blanch for a long time (Table 4) [9,12,13].

In 2009, a study assessed Pb concentrations in wheat, rye/wheat flours, rye flours, rye breads, wheat breads and noodles in Lublin province. The average ranges of Pb content were: wheat (0.057–0.067 mg/kg), rye flours (0.060 mg/kg), breads (0.032 mg/kg), noodles (0.040–0.090 mg/kg) and potatoes (0.027 mg/kg) [9,41]. In 2011, a second study assessed the Pb content in cereal products (flour, groats, bakery products, pasta, flakes and rice) from Podlaskie Voivodeship. These analyses revealed a significantly higher Pb content in buckwheat groats (0.144 mg/kg), bran (0.130 mg/kg) and crunchy bread (0.124 mg/kg) compared to other cereal products. In 10% of the examined buckwheat groats samples, the Pb content was 0.375 mg/kg [42]. Other research has found no concentrations of Pb that exceeded standards in tested milk, herb and spice samples and water, irrespective of the place of origin [40]. Other studies have found lead content in honey to vary by region (ranging from 0.32–2.36 mg/kg) [9].

Research carried out in 2011 found concentrations of Pb in bovine meat tissue of 0.110 mg/kg, while those in other samples taken from pigs and poultry were lower (0.066 mg/kg and 0.034 mg/kg). The livers of the tested animals were characterized by a slightly higher content of Pb in comparison to meat. Pb accumulation in fresh fish was lower (0.010 mg/kg) than in canned fish (0.036 mg/kg). The researchers reported that milk and eggs were characterized by the lowest Pb concentrations. Research found that food products purchased in the commercial network in the Podkarpackie Voivodship did not exceed admissible Pb limits [43].

The literature suggests relatively safe levels of lead in the Polish food supply, but the Pb content should be constantly monitored in food products [9,40,44,45].

### 2.4. Prevention and Monitoring of Lead Poisoning

There is no threshold below which there are no adverse effects of lead exposure; no level of lead exposure is acceptable. Therefore, programs and policies are needed to prevent exposure. Primary prevention requires programs and policies that ensure that all homes, workplaces and the external environment are safe and do not contribute to lead exposure for children or adults. Every effort should be made to increase awareness of the risks of lead and to promote lead-focused nutritional interventions; these are key elements of an effective prevention policy [25].

In view of the high risk to the fetus, specific guidelines for prenatal healthcare providers and women of childbearing age are needed; there are currently no national recommendations or guidelines from maternity, family, pediatric or nursing groups that include lead risk assessment and management of pregnancy and lactation [25].

To prevent exposure of children to lead, children’s hands should be washed frequently, and their intake of calcium and iron optimized, which will reduce the absorption of lead by the body. Children should be discouraged from touching the mouth frequently. Also, public health professionals recommend that homes be kept free of Pb (by cleaning only with cold water), i.e., items that contain lead, such as blinds and jewelry, should be removed from the home. Old water pipes should be replaced; hot water passing through old pipes leads to faster corrosion and the subsequent release of more lead compounds. Public health specialists assume that hot water contains a higher level of lead than cold water, so it is recommended that in the household, especially when cooking and cleaning, mainly cold water be used [2]. The lead content in water may differ based on several factors: water temperature acidity or alkalinity, type and amount of minerals in the water, degree of pipe wear, the time the water remains in the pipes, environmental pollution in the region and the presence of protective scales or coatings in the pipes [5]. However, there is no information in the literature about the difference in lead content in hot and cold water, as this depends on many factors.

Monitoring of lead exposure is needed to assess the magnitude and nature of lead exposure, and to ensure that timely and effective treatment for lead poisoning occurs. International guidelines recommend interventions regarding blood lead concentrations of ≥ 5 μg/dL in pregnant women. The susceptibility of the developing fetus to the adverse effects of lead, and the potential for preventing additional postnatal exposure, justify intervention for pregnant women showing signs of exposure to lead [18]. When mothers’ blood lead concentrations are high, they should be encouraged to pump and eject milk until the level of lead in the blood drops to below 40 μg/dL [25]. In adults, the most critical systems are the kidneys and hematopoietic systems, and peripheral, nervous and circulatory systems. The early effects of lead on these systems and organs appear when the lead concentration in blood is around 300 μg/L, or even below this value. In children, the nervous system is very vulnerable to the negative effects of lead accumulation [15]. The WHO recommends a blood Pb concentration of 5 μg/dL or less for children; damage from lead exposure can occur at levels even below this value [24].

According to the WHO, 98% of adults affected by Pb live in low- and middle-income countries. Workers with blood lead concentrations above 400 μg/L should receive medical care. Exposure to lead should be carefully monitored because the effects of its accumulation in the body are not necessarily immediately visible, and can manifest years later, posing a serious challenge to medicine and public health [16,46].

In most industrialized countries, blood Pb levels are steadily falling due to the cessation of the use of lead fuels. However, some populations are still exposed to high levels of lead, mainly from deteriorating dwellings with residual lead paint [16,19,46,47].

In Poland, lead poisoning is rare, and acute poisoning accounts for a small percentage of all Pb poisoning. Treatment usually takes place in specialized centers, and the most commonly used approach is oral or intravenous chelation using EDTA (a capture compound that removes harmful metals from the body). If indicated, a gastric lavage with 3% sodium sulfate with activated carbon may be used. In cases of encephalopathy and acute or chronic renal failure, hemodialysis is indicated. If intestinal colic occurs, opioids (e.g., codeine) are used. When severe intestinal colic occurs, it is recommended that the opiate drug parasympatykolitycznym be administered [1].

The Polish national guidelines on environmental pollution are in line with those enforced by the European Union. The Polish guidelines cover the scope of initial and periodic medical examinations and contraindications for possible use in the metallurgical sector, where there may be a risk of exposure [15]. There are currently no active Polish preventive programs regarding lead exposure in the workplace. The unit most involved in this type of monitoring is the Occupational Medicine Institute in Łódź (Poland). The agency carries out occupational exposure assessments in workplaces, usually at the request of the employer or local epidemiological and sanitary station. The quality systems used are in accordance with the European standard EN ISO/IEC 17,025, and are regularly verified by international assay quality control systems organized by the German External Quality Assessment Scheme for Toxicological Analyzes in Biological Materials (G-EQUAS-Germany) and United Kingdom National External Quality Assessment Scheme (UK NQAS)—Wolfson Laboratory (United Kingdom). The last occupational lead exposure prevention program was active from 2002–2005 in the Wielkopolska region (Poznań), where batteries were being produced. The goals of the program included reducing the concentration of lead in the blood through health promotion in the workplace, individual (avoiding facial hair, long hair) and collective training (smoking cessation, putting on disposable work clothing), as well as hygiene and workplace procedures (the use of dust masks, washing hands regularly, thorough body washing after work, frequent hydration of the body, avoiding fasting, eating outside the production hall, using disposable dishes). The program had significant benefits [48]. In 2012–2013, the project “Let’s protect children from lead and cadmium—exposure assessment of preschoolers from Piekary Śląskie” (southern part of Poland, very industrialized) was implemented, which included medical examinations and taking blood samples. Less than 8% of the examined children were found to have elevated serum Pb (> 5 μg/dL) and were referred for special treatment [49].

## 3. Conclusions

Any exposure to Pb causes a wide range of physiological, biochemical and behavioral effects. The most dangerous effects occur in the central and peripheral nervous systems, the hematopoietic system, the cardiovascular system and in some organs such as the liver and kidneys. A review of the literature suggests that some regions in Poland are more contaminated with lead, both in soil and in the air, than others. Public policy should have as its goal the minimization of lead exposure in the workplace, home, and the larger human environment. Therefore, blood Pb concentrations should be monitored in all groups, i.e., not only in women planning pregnancy, but also in young children, adults and the elderly. A nation-wide lead monitoring program is recommended. Social and professional education should promote lead awareness, increase health knowledge and provide the skills necessary to prevent lead poisoning. The Polish food supply appears to be safe. A comprehensive regulatory plan and health intervention program should be developed and implemented in Poland to reduce lead exposure and associated health risks, e.g., cardiovascular disease, fertility disorders, kidneys diseases and central nervous system diseases.

## Figures and Tables

**Figure 1 ijerph-17-04385-f001:**
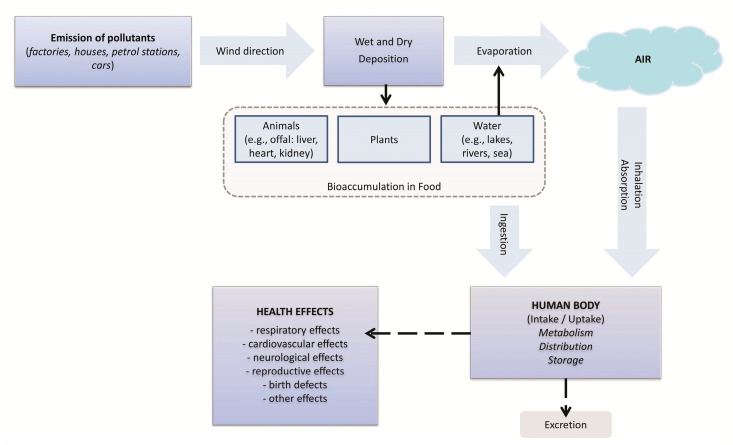
Exposure to lead pollution and possible health effects in humans (Adopted from: [2]).

**Figure 2 ijerph-17-04385-f002:**
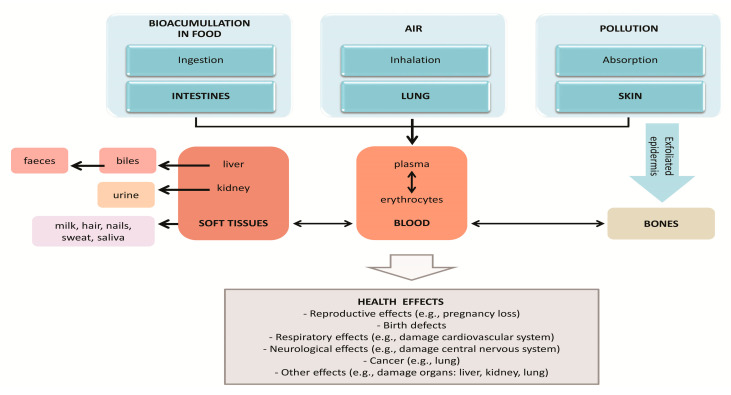
Absorption, distribution and excretion of Pb from the body (thin arrows—absorption paths; bold arrows—excreting pathways; dashed arrows—movement inside the body. The main locations of Pb in the body (blood, bones, soft tissues) [4,18,19].

**Figure 3 ijerph-17-04385-f003:**
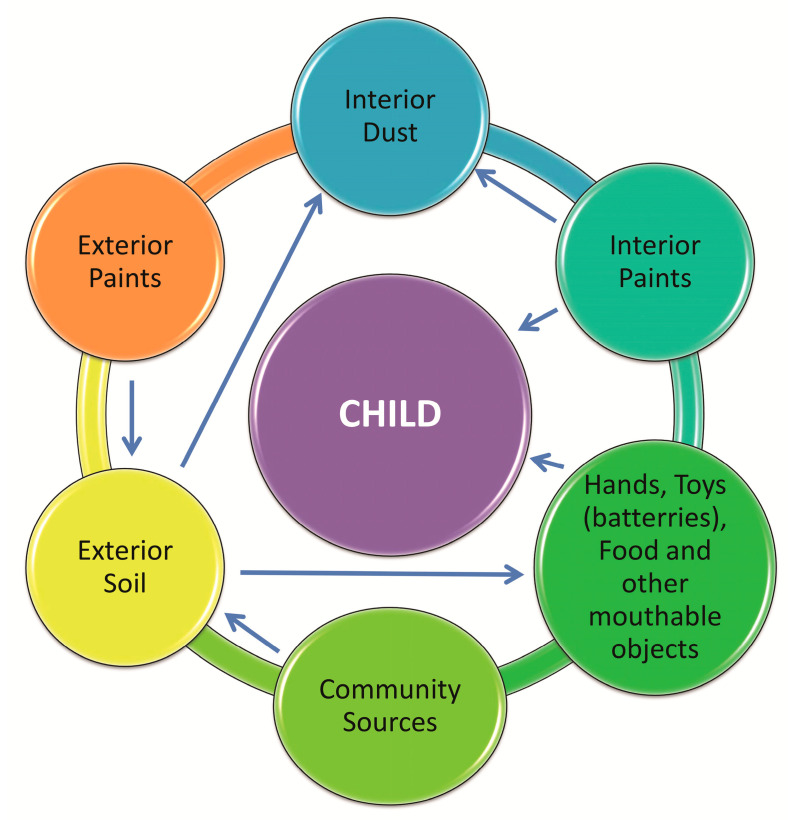
Possible sources of lead poisoning among children in the home environment (Adopted from: [2]).

**Table 1 ijerph-17-04385-t001:** Symptoms of poisoning according to the degree of exposure in children and adults (Adopted from [4]).

Blood Lead Level (µg/L)	Degrees of Lead Poisoning	Symptoms	
		Adults	Children
<10	low	passing through the placenta into the fetal bloodstream	IQ reduction, learning and memory disabilities, growth disorders, reduced development, motor coordination, hearing, speech and verbal skills, symptoms of hyperactivity
10–40	mild	elevated blood pressure, slowed nerve conduction	deceleration of nerve conduction and vitamin D metabolism, impaired hemoglobin synthesis, sporadic intestinal discomfort, muscle pain, irritability, fatigue, apathy
40–70	moderate	drowsiness, fatigue, mood swings, reduced mental abilities, impaired fertility, chronic hypertension, impaired hemoglobin synthesis	difficulty concentrating, trembling, fatigue, muscular weakness, headache, vomiting, constipation, weight loss
70–100	serious	metallic taste in the mouth, constipation, headaches, abdominal pain, insomnia, memory loss, decreased sex drive, nephropathy	colic (severe intestinal musculoskeletal contractions), lead limbs (dark teeth and/or gums), anemia, nephropathy, encephalopathy, paralysis
>100	acute poisoning	encephalopathy, anemia, death (> 150 µg/L)	convulsions, death (usually below 150 µg/L)

**Table 2 ijerph-17-04385-t002:** The effects of Lead—a summary.

Lead’s Effects	Summary	
Metabolic and Genetic Effects	Impairs multiple biochemical processes	
	Interacts with biological molecules	
	Reduces the activity of various enzymes	
	Causes oxidative stress	
	Causes gene expression alterations	
	Inhibits the enzyme ferrochelatase	
	Can causes rapid depletion of antioxidants in the body	
	Can increase the production of reactive oxygen, as well as reactive forms of nitrogen	
Location of Lead in the Body	Most lead is stored in the bones	
	Bone-to-blood lead mobilization increases during advanced age	
	Calcium deficiency is exacerbated, bone-to-blood lead mobilization	
Children	Adversely affects the central nervous system and development of children	
	Can interfere with the ability to learn, impair memory, lower IQ and interfere with growth and development	
	Can affect speech and hearing, cause hyperactivity, nerve conduction, intestinal discomfort, constipation	
	Vomiting, weight loss muscle aches	
	Can lead to anemia, nephropathy, paralysis, convulsions or death	
	Damage can begin as early as pregnancy	
	Increases the risk of miscarriage, stillbirth, low birth weight and underdeveloped children	
	Children are lead exposed via inhalation and ingestion	
	Increased risk occurs in families where one parent works in an environment where high levels of lead are present	
Adults	Can result in serious morphological and functional changes in some organs	
	Can cause changes in the nervous system, the circulatory system, the gastrointestinal system	
	Hormonal, astringency of the mouth, metallic taste in the mouth, and thirst or death	
	Affects myocytes of the muscular layer of blood vessels	
	Can result in an increase in blood pressure (not firmly established)	
	Can adversely affect blood and the metabolism of blood cells	
	Can have adverse effects on the nervous system, fertility, miscarriages	

**Table 3 ijerph-17-04385-t003:** Threshold lead contents in selected food products [9,12,39,40].

Food Product	Threshold Content (mg/kg Fresh Matter)
Milk	0.02
Meat (bovine animals, sheep, pig, poultry)	0.1
Offal (bovine animals, sheep, pig, poultry)	0.5
Fish	0.3
Cereal	0.2
Vegetable (leaf, fresh herbs, fungi, seaweed)	0.1–0.3
Potatoes	0.1
Fruits	0.1–0.2
Fat and oil	0.1
Honey	0.1
Drinking water	0.1*

* 10 µg/L—Regulation of the Minister of Health on the quality of water intended for human consumption of December 7, 2017 (Journal of Laws of 2017, item 2294). Polish regulations.

**Table 4 ijerph-17-04385-t004:** Lead exposure in Poland—a summary.

Lead’s Exposure	Summary
In the environment	Poland has the richest Pb deposits in Europe
	Pb has been widely used since the 12th Century
	Lead in gasoline was a major pollutant for much of the 20th Century
	The south-west regions of the country are the most polluted, caused by higher levels of industrial activity and transboundary transport
	The most polluted air occurs in the vicinity of mines and metalworks
	Pb generally ranks first as a pollutant in waste management, second in industries burning coal, and third in road transport
	The work environment is a significant source of Pb exposure
In paints	Present in paints manufactured before 1978
	Can still be found on many painted surfaces in older homes
	Repainting and renovations can create a serious problem with lead dust
	Old lead paint can contaminate the soil
In food	Nearly all agricultural soils are characterized by some natural Pb
	Lead absorption by plants depends on the properties of the soil, the characteristic features of the species and the physiological state of the plant
	An important source of contamination of food products can be technological processes and devices used in food production
	Pb contamination depends on the infrastructure of the area and industry

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
