# Peer review of "Lead Toxicity and Pollution in Poland"

_ijerph, 2020, doi:10.3390/ijerph17124385_

Round 1
Reviewer 1 Report
Dear Authors.
The title “The Lead Toxicity and Pollution with this Element in Poland” correspond well with the text. However the “lead toxicity” part was already described in a review in 2016 (https://doi.org/10.1515/intox-2015-0009) – reference no.2. The new and important contribution of this review gives inside on the pollution of lead in Poland.
Section 2.1. Is the chemical composition of the lead compound (organic, inorganic) important? Is there any difference in the absorption? It is of importance to describe which mechanism of absorption causes more negative effects.
Section 2.2.2. There is a lack of information about the accumulation of Pb in adipose tissue and the transfer of Pb to the blood after weight loss (https://doi.org/10.1016/j.scitotenv.2020.137458).
Section 2.4.3. The Pb contamination in food on the polish market is the essential part of this review but there are not many values. In line 294 is stated: “Among food products, the highest Pb concentrations have been observed in vegetables, potatoes and milk” but there are no values given - is it above the legal threshold (what is the legal threshold in Poland?). Line 305 “potatoes (0.027 mg/kg)” and line 312 „bovine meat tissue of 0.110 mg/kg” – there is more lead in meat than potatoes but in line 294 it is not stated - (recommendation: compare levels in table). Line 316 “milk and eggs were characterized by the smallest Pb level” – again not consistent with line 295.
In section 2.5. I am missing the part about lead poisoning treatment. Is there any data considering the cases of lead poisoning and/or treatment procedures in Poland? In lines 336 and 340, the author recommended to clean household items only with cold water but there is no explanation why this is crucial. The author stated in line 338: “Public health professionals assume that hot water contains a higher level of lead than cold water” again no explanation why or how much higher is the lead content in hot water compared with cold water.
In the conclusion, the authors recommend “A nation-wide lead monitoring program is indicated” (line 371). Is there today any national guideline according the lead pollution in the environment and/or food in Poland? This should be described in the section 2.5. with the prevention program in Poland (if there is any).
Detected typos and errors in the text (numbers represents lines in the text):
-85 the sentence starts with a different font;
-120 “enzymethusle” this term is to me unknown;
-209 “but it is not this fact is” – but this fact is not;
-258 “for us” – for the human health;
-293-294 “lettuce” is twice written;
-310, 319, 336 – lead is written PB instead of Pb;
-336 there is only a closing parenthesis: “by cleaning only cold water) and” - by cleaning only WITH cold water;
- 346 double space between the coma sign and “they”;
-351 “fretless”- in this sentence it refers to the threshold limit (non-threshold). The whole sentence should be reformulated;
-359 “levels of blood Pb in are steadily falling”;
-376 “[…] reproduction, kidneys, central nervous system.” Disease? Damage?
References are not formatted according to the guidelines (especially the authors names). On the line 421 is an additional “enter”. The WHO AQGs, line 446, was published in 2015 or 2016? In Citation number 22., 25., 27., 31., 38. and 39. are “et al.” insertions is after 3 names. In previous publications in IJERPH “et al.” was used if there was more than 10 authors. Reference no. 28. is incorrectly written and no. 40 has a colon instead a dot after the last name.
The literature used is up-to-date: 19 of 40 publications are from the last 5 years. One publication is from the year 1993 and 19 publications from the years 2005-2014.

Author Response
Replies to the comments of the Reviewers
We agree with the comments raised by the Reviewers.
In the revised manuscript we have introduced appropriate corrections (in blue).
Reviewer report (1):
Thank You very much for Your review.
Replies to Editor’s comment:
- The title “The Lead Toxicity and Pollution with this Element in Poland” correspond well with the text. However the “lead toxicity” part was already described in a review in 2016 (https://doi.org/10.1515/intox-2015-0009) – reference no.2. The new and important contribution of this review gives inside on the pollution of lead in Poland. – thank you for Your comment
- Section 2.1. Is the chemical composition of the lead compound (organic, inorganic) important? Is there any difference in the absorption? It is of importance to describe which mechanism of absorption causes more negative effects. - has been changed, some of them removed and added in the Introduction - line 33-34, 35-37, 44-48
- Section 2.2.2. There is a lack of information about the accumulation of Pb in adipose tissue and the transfer of Pb to the blood after weight loss (https://doi.org/10.1016/j.scitotenv.2020.137458). - has been added - line 125-126
- Section 2.4.3. The Pb contamination in food on the polish market is the essential part of this review but there are not many values. In line 294 is stated: “Among food products, the highest Pb concentrations have been observed in vegetables, potatoes and milk” but there are no values given - is it above the legal threshold (what is the legal threshold in Poland?). Line 305 “potatoes (0.027 mg/kg)” and line 312 „bovine meat tissue of 0.110 mg/kg” – there is more lead in meat than potatoes but in line 294 it is not stated - (recommendation: compare levels in table). Line 316 “milk and eggs were characterized by the smallest Pb level” – again not consistent with line 295. - has been changed – line 267-299, 302-304 and added Table 3 represent “Threshold lead contents in selected food products” according to suggestion
- In section 2.5. I am missing the part about lead poisoning treatment. Is there any data considering the cases of lead poisoning and/or treatment procedures in Poland? – has been added – line 384-390
In lines 336 and 340, the author recommended to clean household items only with cold water but there is no explanation why this is crucial. The author stated in line 338: “Public health professionals assume that hot water contains a higher level of lead than cold water” again no explanation why or how much higher is the lead content in hot water compared with cold water. - has been changed – line 354-362
- In the conclusion, the authors recommend “A nation-wide lead monitoring program is indicated” (line 371). Is there today any national guideline according the lead pollution in the environment and/or food in Poland? This should be described in the section 2.5. with the prevention program in Poland (if there is any). – has been added – line 391-413
- Detected typos and errors in the text (numbers represents lines in the text):
-85 the sentence starts with a different font; - has been changed – line 76-77
-120 “enzymethusle” this term is to me unknown; - has been changed – line 110
-209 “but it is not this fact is” – but this fact is not; - has been changed – line 202
-258 “for us” – for the human health; - has been changed – line 264
-293-294 “lettuce” is twice written; - has been changed – line 296
-310, 319, 336 – lead is written PB instead of Pb; - has been changed – line 39, 255, 320, 330, 339, 352, 422
-336 there is only a closing parenthesis: “by cleaning only cold water) and” - by cleaning only WITH cold water; - has been changed – line 352
- 346 double space between the coma sign and “they”; - has been changed – line 368
-351 “fretless”- in this sentence it refers to the threshold limit (non-threshold). The whole sentence should be reformulated; - has been changed – line 372-373
-359 “levels of blood Pb in are steadily falling”; - has been changed – line 381
-376 “[…] reproduction, kidneys, central nervous system.” Disease? Damage? - has been changed – line 429
- References are not formatted according to the guidelines (especially the authors names). On the line 421 is an additional “enter”. The WHO AQGs, line 446, was published in 2015 or 2016? - has been changed, citation number: 10 (line: 473-74), 24 (line: 507-508)
In Citation number 22., 25., 27., 31., 38. and 39. are “et al.” insertions is after 3 names. In previous publications in IJERPH “et al.” was used if there was more than 10 authors. Reference no. 28. is incorrectly written and no. 40 has a colon instead a dot after the last name. - has been changed, citation number: 18 (line: 493-95), 28 (line: 521-23), 30 (line: 527-29), 38 (line: 546-48), 46 (line: 563-565), 19 (line: 496-98)
- The literature used is up-to-date: 19 of 40 publications are from the last 5 years. One publication is from the year 1993 and 19 publications from the years 2005-2014. – we checked again, refreshed and added some new publications: 21, 23, 27,31-34, 39, 48, 49
We are grateful to the Reviewers for all insightful suggestions.
We appreciate the time and efforts by the Editor and Reviewers in reviewing this manuscript.
We have addressed issues indicated in the review report, and We believed that the revised version manuscript can meet the Journal publication requirements.
Reviewer 2 Report
A review of an important topic but has at an academic and unscientific level. A greater critical discussion is needed. The references reported are very few and do not allow critically discuss the topic. If it is a scientific article, it hardly needs references from just one country, a wider papers research is certainly needed for discussion.In some parts the paper appears a bachelor thesis (i.e. section 2.1-2.1.1). The authors need to review the manuscript in this direction.
In the introduction explain why it is important to study lead in poland.
Line 50 remove "results" these are not results.
Use lead or Pb in the manuscript.
section 2.2: a summary table of the reported informations, could improve the manuscript
section 2.3: a summary table of the reported informations, could improve the manuscript
after these changes I think the paper can be published
Author Response
Replies to the comments of the Reviewers
We agree with the comments raised by the Reviewers.
In the revised manuscript we have introduced appropriate corrections (in blue).
Reviewer report (2):
Thank You for Your specific remarks.
As suggested by the Reviewer, we made amendments:
- A review of an important topic but has at an academic and unscientific level. A greater critical discussion is needed. The references reported are very few and do not allow critically discuss the topic. If it is a scientific article, it hardly needs references from just one country, a wider papers research is certainly needed for discussion. - some of them added in the text - line 125-126, 146-147, 153-155, 168-170, 205-206, 211-215, 234-236
- In some parts the paper appears a bachelor thesis (i.e. section 2.1-2.1.1). The authors need to review the manuscript in this direction. - has been changed, some of them removed and added the Introduction - line 33-34, 36-38
- In the introduction explain why it is important to study lead in Poland. - has been changed – line 44-48
- Line 50 remove "results" these are not results. - has been changed – line: 54
- Use lead or Pb in the manuscript. - has been changed – line: 23, 24, 109, 110, 171, 274, 277
- section 2.2: a summary table of the reported informations, could improve the manuscript - has been added Table 2 – line: 238-241
- section 2.3: a summary table of the reported informations, could improve the manuscript has been added Table 4 – line: 335-336
We are grateful to the Reviewers for all insightful suggestions.
We appreciate the time and efforts by the Editor and Reviewers in reviewing this manuscript.
We have addressed issues indicated in the review report, and We believed that the revised version manuscript can meet the Journal publication requirements.
Round 2
Reviewer 2 Report
This revised version has improved greatly.
Only one point: I suggest to modify heavy metals in Potential toxic elements
Author Response
Replies to the comment of the Reviewer
We agree with the comments raised by the Reviewers.
In the revised manuscript we have introduced appropriate corrections (yellow).
Thank You for Your specific remarks.
As suggested by the Reviewer, we made amendments:
Only one point: I suggest to modify heavy metals in Potential toxic elements - has been changed – line 12, 27, 45,167, 260-261.
We are grateful to the Reviewers for all insightful suggestions.
We appreciate the time and efforts by the Editor and Reviewers in reviewing this manuscript.
We have addressed issues indicated in the review report, and We believed that the revised version manuscript can meet the Journal publication requirements.
